# Analysis of the Relationship between Higher-Order Factor Structure of Personality Disorders and the Five-Factor Model of Personality

**DOI:** 10.3390/brainsci13040605

**Published:** 2023-04-03

**Authors:** Danilo Pešić, Dušica Lečić-Toševski, Marko Kalanj, Ivan Ristić, Olivera Vuković, Bojana Pejušković

**Affiliations:** 1Clinic for Children and Adolescence, Institute of Mental Health, 11000 Belgrade, Serbia; 2School of Medicine, University of Belgrade, 11000 Belgrade, Serbia; 3Serbian Academy of Sciences and Arts, 11000 Belgrade, Serbia; 4Clinical Department for Psychotic Disorders, Institute of Mental Health, 11000 Belgrade, Serbia; 5Department for Research and Education, Institute of Mental Health, 11000 Belgrade, Serbia; 6Clinical Department for Crisis intervention and Affective Disorders, Head, Institute of Mental Health, 11000 Belgrade, Serbia

**Keywords:** personality disorder, ICD-11 classification, five factor model of personality, metatraits, broad personality disorder factor, canonical analysis of covariance

## Abstract

The growing body of evidence on the dimensional classification of personality disorders (PD) has resulted in its acceptance in the ICD-11 classification, which abolished categories and retains only a general description of PD. Specifying the type of PD is optional, and the suggested domains represent maladaptive variants of the five-factor model of personality (FFM). The aim of our study was to explore the existence of a joint structure between maladaptive and normal personality traits, and to investigate how these structures are integrated. The study included 223 patients who had been diagnosed with PD and completed the Structured Clinical Interview for DSM-5 Personality Disorders and the NEO Personality Inventory-Revised (NEO-PI-R). To determine the degree of overlap between PD domains and NEO PI–R scales, a canonical analysis of covariance was conducted. Our findings showed a relationship between the internalizing PD spectrum (consisting of avoidant, dependent, and borderline traits with detached and anankastic traits) and high neuroticism, low conscientiousness, and moderately low agreeableness and extroversion, suggesting the existence of a broad personality disorder factor. However, the internalizing dimensions exhibited a more pronounced effect within this construct. Furthermore, we identified a second function that demonstrated a link between the externalizing PD spectrum (including narcissistic, histrionic, and antisocial traits) and high extraversion, high openness, and low agreeableness, suggesting the existence of an externalizing factor. Overall, our findings provide evidence for a joint structure of maladaptive and normal personality traits in a sample of personality disorders and emphasize the importance of integrating the FFM model in PD evaluation in clinical practice, suggesting that differentiating between major subgroups could assist in adjusting therapeutic approaches.

## 1. Introduction

A growing body of research suggests that personality disorders (PD) should be diagnosed dimensionally, and integrating a dimensional model of PD with a general personality structure can provide practical and conceptual advantages [1,2].

The five-factor model (FFM) stands out as the most comprehensive model of the general personality structure, and the basic dimensions that exist in other inventories of personality can be fully captured by the FFM [3]. The NEO-Personality Inventory-Revised (NEO-PI-R) is the most commonly used assessment tool for the FFM and consists of five personality domains: Neuroticism (N), extraversion (E), openness to experience (O), agreeableness (A), and conscientiousness [4]. Studies also indicated that PD can be well described by FFM, which is widely regarded as representative of the structure of both normal and abnormal personality traits [5].

The use of personality traits in diagnosing PD has a long history in the existing literature and has been widely discussed, particularly from the perspective of the FFM [6]. Results from research studies suggest that both FFM counts and the full proto-type-matching technique, such as FFM similarity scores, are equally effective in relating to PD symptoms [7].

The correlation between the Five-Factor Model (FFM) and pathological personality traits has been demonstrated across different cultures and using various instruments [8,9]. In a study exploring the correlation between pathological personality traits and the Five-Factor Model (FFM) across different cultures and instruments, researchers used the Chinese Adjective Descriptors of Personality (CADP) to measure normal personality traits comparable to the FFM. They found significant links between all five CADP traits and nearly all 11 PD functioning styles in both patients and healthy volunteers [8].

The DSM-5 Alternative Model of Personality Disorder (AMPD) and ICD-11 classification of PD both demonstrate alignment with each other and with the “Big Five” personality model. It has been established by numerous studies that the domains in both models represent maladaptive variants of the Big Five [10,11,12]. Notably, even criterion A in AMPD, which deals with self–other deficit, can be regarded as a maladaptive variant of the FFM [13]. Empirical confirmation of agreement with the FFM is evident in ICD-11 domains, with four out of the five Big Five traits matching up. Negative affectivity is aligned with N, detachment with low E, dissociality with low A. Disinhibition and anankastia, as opposing domains, align with opposite ends of conscientiousness. However, it is worth noting that ICD-11 domains lack a counterpart for openness to experience [14,15].

The significance of hierarchical models in comprehending both normal and abnormal personality structures has been increasingly acknowledged in recent years [16].

For instance, different “factor solutions” may result in varying numbers and compositions of factors. The Personality Psychopathology Five (PSY-5) analyses, for example, identified five factors that correspond to the domains in the AMPD. However, other factor solutions may result in a varying number and composition of factors and may collapse psychoticism into other factors, resulting in the identification of the “pathological Big Four”, “Big Three”, or a two-factor solution of internalizing and externalizing factors. In addition, there is a one-factor solution that describes overall personality pathology [17].

Similarly, research on the FFM of personality has shown that its traits are not independent but rather correlated, indicating the presence of metatraits [18,19]. The metatraits of “Alpha/Stability” and “Beta/Plasticity” were initially identified and later criticized [20,21,22], reevaluated and validated [23,24,25] and a recent meta-analysis has confirmed their existence [26]. Strus et al. expanded this model by introducing the Circumplex of Personality Metatraits (CPM), which includes also Gamma and Delta metatraits. The CPM is a circular model with four bipolar axes of personality metatraits representing combinations of the Big Five, providing a unified framework for understanding both pathological and non-pathological functioning. These metatraits can represent tendencies towards socialization and mental health, or the opposite predisposition towards PD and behavioral dyscontrol [27,28].

Over and above, recent research has provided compelling evidence that a single integrative hierarchy is the most accurate and comprehensive way to conceptualize both normal and abnormal personality traits [16]. This perspective has been supported by research utilizing joint factor analysis of instruments designed to measure normal and pathological traits. Kajonius’ research on a Swedish community sample suggests that the FFM and the Personality Inventory for DSM-5 (PID-5) may be interchangeable, given their mutual structure [29].

This study builds upon our previous research [30], where we aimed to cross-validate the proposed ICD-11 domains in a sample of PD-diagnosed patients, based on Mulder et al.’s work [31]. In a previous study, we found partial replication of the five domain structures, with the strongest evidence supporting the existence of the first two factors. These factors were labeled as borderline internalizing, which included items related to borderline, avoidant, and dependent traits, and disinhibited/borderline externalizing factors, which included items related to narcissistic and histrionic traits. However, the remaining three factors in our study, labeled as antisocial, anankastic, and detached, showed significantly less robustness compared to the first two factors [30].

The primary objective of this study was to investigate the relationship between higher-order factors, which we previously identified in the clinical population of PD [30], and the FFM traits. We used canonical analysis of covariance (CAC) to accomplish this and also aimed to identify distinctive PD patterns. Our main research question focused on the extent and manner in which maladaptive and common trait structures are integrated. We hypothesized that there would be evidence of a mutual structure of a latent higher-order factor.

## 2. Materials and Methods

### 2.1. Sample

The study was carried out at the Institute of Mental Health, Belgrade, from January 2011 to June 2016. The sample consisted of 223 patients diagnosed with PD according to ICD-10 criteria. Out of 223 patients diagnosed with PD, 112 were recruited from day hospital for affective disorders and day hospital for adolescents (18 or more years of age), while 111 were inpatients at the clinical ward for affective disorders. Experienced clinicians confirmed the prior diagnosis for all participants. Exclusion criteria were the presence of an organic mental disorder, mental retardation, psychotic disorder, severe substance and drug abuse, severe major depressive disorder, manic episode, and bipolar disorder.

### 2.2. Study Design and Statistical Framework

This paper is the second phase of our research project [31]. In the first phase, we attempted to cross-validate the study conducted by Mulder and colleagues, who validated five proposed domains of PD for the new ICD-11 classification [31]. Our primary goal was to use the same framework of grouping and choosing DSM items and to perform the same procedure of grouping, but on a different population, specifically a sample of primary PD patients. We applied the same framework with a slightly smaller number of item parcels and a slightly different grouping of the DSM symptom criteria in a sample of PD patients. To analyze the data, we used exploratory factor analysis (EFA) with principal axis factoring as the extraction method and Promax as the oblique rotation method. Exploratory factor analysis (EFA) was conducted on 21 item parcels, with a ratio of 10.6 cases per variable. The Kaiser–Meyer–Olkin measure of sampling adequacy was 0.81, and Bartlett’s test of sphericity was significant ((210) = 1350.62, *p* < 0.01), indicating that the data were appropriate for factor analysis. We extracted five factors with eigenvalues greater than 1, which collectively accounted for 41.39% of the variance. Factor score estimates were generated for each of the five factors by summing the individual items within each item parcel that loaded onto the respective factor, and factor reliabilities were assessed using Cronbach’s alpha. Reliability estimates were strong for the borderline-internalizing (α = 0.86) and antisocial (α = 0.80) factors, acceptable for the disinhibited/borderline externalizing factor (α = 0.71), and poor for the anankastic (α = 0.55) and detached (α = 0.53) factors [30]. 

Our study partially confirmed the five domain structures proposed by Mulder et al. [13], with the borderline symptoms, along with avoidant and dependent symptoms, forming the borderline internalizing factor, and narcissistic and histrionic symptoms forming the disinhibited/borderline externalizing factor. However, the other three factors identified in our study, namely antisocial, anankastic, and detached, were less robust. For more details, please refer to our published paper [30].

To assess the relationship between higher-order PD domains and NEO PI–R scales, using a simple correlation would not be the most accurate approach since it assumes the mutual independence of variables within the same set. Canonical correlation analysis (CCA) is a method used to analyze the associations between two multivariate groups, such as the PD set and the FFM set. CCA works by finding linear combinations of each group that maximize the Pearson correlation coefficient between them.

As a result, it can also serve as a dimension reduction method, as each multidimensional variable is reduced to a linear combination [32]. Since CCA examines the correlation between a synthetic criterion and a synthetic predictor variable, which are weighted based on the relationships between the variables within the sets, it can be conceptualized as a simple bivariate correlation (Pearson r) between the two synthetic variables [33].

While CCA is a powerful technique, it has some limitations, such as the assumptions of multivariate normality and strict regularity of intercorrelation matrices, sensitivity to outliers, and the need for a relatively large sample size for stable parameter estimation [34,35]. Therefore, in this study, we utilized a different approach called canonical analysis of covariance (CAC), also known as quasi-canonical analysis [36,37]. It maximizes the covariance between, not necessarily orthogonal linear combinations of two sets of variables. The benefits of CAC include its lack of assumption of multivariate normality, insensitivity to the regularity of intercorrelation matrices, less sensitivity to outliers, and applicability to smaller sample sizes due to the stability of its estimated parameters and the results of its significance tests [38]. Given these advantages, we employed CAC to assess the association between two sets of interrelated variables, namely the higher-order factors of personality disorders and the domains of the five-factor personality model. The analysis was conducted using an SPSS (version 23.0, Chicago, IL, USA) macro program [38].

### 2.3. Assessment

All participants underwent assessment using the Structured Clinical Interview for DSM-5 Personality Disorders [39], the Structured Clinical Interview for DSM-5 (SCID-5) for making the major DSM-5 diagnoses [40], as well as completing the self-report NEO Personality Inventory-Revised (NEO PI-R) and a socio-demographic questionnaire. The NEO PI-R is a measure of the Five-Factor Model of personality and consists of 240 items rated on a Likert scale ranging from 1 (strongly disagree) to 5 (strongly agree). The questionnaire was standardized for the Serbian population, making it an appropriate tool for this study [41]. The assessments were conducted during the patients’ clinical remission of their comorbid disorders, at the endpoint of their inpatient stay, when both the patients’ and clinicians’ impressions of symptom reduction and well-being matched. 

## 3. Results

In this study, a total of 223 participants (66.8% female) between the ages of 18 and 67 (mean age 37.6 ± 13.0) were included. The mean educational level of the participants was 12.6 ± 2.6 years, with 11.1% having completed primary school, 58.0% having finished high school, and 30.9% being university students, graduates, or post-graduates. The most commonly diagnosed ICD-10 disorder was emotionally unstable PD, accounting for 65.1% of diagnoses, followed by unspecified PD (9.7%). Among the comorbid mental disorders, mood disorders were the most prevalent (58.2%), followed by anxiety disorders (46.4%) and substance-related disorders (25.0%).

The study’s results regarding the linear correlations between variables from the FFM and previously extracted PD domain factors are presented in Table 1. The table provides information on the strength and direction of the correlations between the five-factor personality model and the various personality disorder domain factors.

The results of the quasi-canonical correlation analysis indicate the presence of two quasi-canonical functions, each comprising pairs of quasi-canonical synthetic variables belonging to the PD set and FFM set, respectively. These pairs of variables are significantly correlated with each other, as evidenced by the strong quasi-canonical correlation coefficient (ρ). The quasi-canonical correlation coefficient represents the correlation between the two synthetic variables in a given quasi-canonical function.

Both the first quasi-canonical function (ρ = 0.70, ρ2 = 0.50, F = 221.35, *p* = 0.000) and the second quasi-canonical function (ρ = 0.63, ρ2 = 0.41, F = 221.35, *p* = 0.000) explain a statistically significant amount of shared variance between the sets of variables. The interpretation of most parameters is similar to those used in CCA. 

The quasi-canonical function coefficients and structure coefficients for the first quasi-canonical function are presented in Table 2.

All variables in the PD set were found to have significant contributions to the first synthetic variable, which accounts for 40.4% of the variance within the variable set. The borderline internalizing (composed of avoidant, dependent, and borderline items), detached, and anankastic domains were the most relevant, as indicated by higher function and structure coefficients, while the disinhibited externalizing and antisocial domains made secondary contributions.

As for the variables in the FFM set, the second synthetic variable accounts for 39.2% of the variance and includes neuroticism (N+), negative conscientiousness (C-), moderately negative agreeableness (A-), and moderately negative extraversion (E-), with negative openness (O-) having no significant contribution. This was determined by lower function and structure coefficients.

Furthermore, redundancy analysis was conducted to explore the relationship between the two sets of variables. The results showed that 19.5% of the variance in the PD variables can be explained by the five-factor personality measures, while 19.1% of the variance in the five-factor personality variables can be explained by the PD variable set. 

The quasi-canonical function coefficients and structure coefficients for the second quasi-canonical function are presented in Table 3.

The synthetic variable in the PD variable set was found to be primarily influenced by the disinhibited/borderline externalizing and antisocial factors, with a relatively weak negative association with the anankastic factor. Collectively, these variables accounted for 21.8% of the variance in that variable set. On the other hand, the second synthetic variable in the FFM set explained 24.9% of the variance and was characterized by positive extraversion (E+), positive openness (O+), and moderately negative agreeableness (A-).

In terms of redundancy analysis, the results were similar to the first function. The proportion of variance explained was relatively low, with 8.4% of the variance in the PD variables being accounted for by the five-factor model variables and 9.4% of the variance in the five-factor model variables being explained by the PD factors.

## 4. Discussion

Clinical experience has indicated that individuals diagnosed with PD exhibit more similarities than differences in terms of the pervasiveness, inflexibility, and clinically significant distress of their symptoms and behaviors. The research focused on personality in this population has also reached the same conclusion, revealing that all individuals with PD tend to exhibit the same configuration of traits, primarily characterized by high levels of neuroticism and low levels of agreeableness and conscientiousness [5].

Joint factor analysis studies investigating FFM alongside measures designed to evaluate abnormal personality typically reveal the existence of shared factors that account for variance in both types of measures [42].

In Kajonijus and Daderman’s study, one of their objectives was to evaluate dispositions for DSM categories based on normal personality continuums. To accomplish this, they connected personality disorder categories to normal personality traits on a continuum. To calculate PD scores, they utilized the FFM-count method, which depends on expert prototypes that assign each PD category a specific set of FFM facets. To create higher-order variables, the researchers employed a “bass-ackwards” approach, which allowed for a top-down structure with path coefficients. They achieved this by developing a hierarchical factor structure level by level, beginning with the extraction of one fixed component and continuing while correlating the components between levels. Due to the known comorbidity between factors, they chose to use EFA. The higher-order factors that were derived from the factor analyses aligned well with previously established higher-order organizational frameworks, such as internalizing and externalizing. This concept is similar to our concept, and these results align with our results [43].

In our study, we used a different power statistic technique that does not assume multivariate normality and is suitable for smaller sample sizes to evaluate the association between higher-order factors of PD and the domains of the FFM.

The first quasi-canonical function in quasi-canonical synthetic variables belonging to the PD set primarily reflects the Borderline-internalizing domain (avoidant, dependent, and borderline traits), while also showing significant loadings from detached and anankastic and to a lesser extent from the other two domains. However, factor loadings for the disinhibited/borderline-externalizing and antisocial domains were markedly lower.

The primary characteristic of the first quasi-canonical function in quasi-canonical synthetic variables belonging to the FFM set is a high level of neuroticism (N+), followed by low conscientiousness (C-), and moderately low levels of agreeableness (A-) and extraversion (E-).

If we analyze the contribution of pathological traits separately, we need to examine the quasi-canonical synthetic variables belonging to the PD set and its structure. By doing so, we can connect our results with the findings of Jahng et al., who conducted a study on individuals with substance use disorder [44]. They used bifactor modeling to uncover a robust general factor of personality disorder (g-PD) and a residual cluster B PD factor.

The g-PD factor can be explained in two ways: firstly, through its strong connection with neuroticism, which is characterized by negative affectivity and emotional dysregulation. Secondly, g-PD is connected to interpersonal dysfunction, as the PDs with the most impaired interpersonal functioning (avoidant, dependent, paranoid, and schizoid) have the highest loading on the g-PD factor. These PDs are characterized by “adaptive failure” as the central problem. This finding is almost identical to our study, as the most significant contributor to the PD variable set is the borderline-internalizing factor, which is strongly related to neuroticism from the FFM set. Additionally, the PD variable set primarily consists of the borderline-internalizing factor (including all avoidant and dependent items, along with all borderline and some paranoid items) and the detachment (schizoid) factor. This has a similar saturation structure to the aforementioned study and contributes to core interpersonal dysfunction.

In Sharp and colleagues’ study, a bifactor analysis revealed that all borderline personality disorder (BPD) criteria loaded onto a single g-PD factor, which corresponds to the disorder in self-experience and interpersonal functioning (Criterion A) in the general definition of PD in ICD-11. Similarly, in our previous work, all BPD items loaded entirely onto one factor, which was the most robust, along with avoidant and dependent items, suggesting that BPD criteria may represent general features of PD, such as problems with identity, self-direction, empathy, and intimacy, as noted by Sharp and colleagues [45]. 

If we analyze the contribution of normal traits separately, we need to examine the quasi-canonical synthetic variables belonging to the FFM set and its structure.

In our analysis of the first quasi-canonical function FFM variable set, high positive neuroticism (N+) and low scores on all other traits, particularly high negative conscientiousness (C-), negative agreeableness (A-), and negative extraversion (E-), were the primary contributors. 

The linear composite of traits in our research (N+, C-, A-, E-) is very similar to the Gamma-Minus/Disharmony, which is positively correlated with N and negatively correlated with all the remaining traits (N+, C-, A-, E-, O-) in the CMP model proposed by Strus and colleagues. This metatrait represents the most maladaptive configuration, characterized by interpersonal problems and a general predisposition to PD [46]. On the other hand, “Gama-Plus/Integration” is negatively correlated with N and positively correlated with all the remaining traits (N-, E+, O+, A+, C+). It represents a tendency towards socialization and mental health and provides a better empirical understanding of the general factor of personality (GFP) from an FFM perspective [27,28].

The first quasi-canonical function encompasses not only borderline PD items but also avoidant and dependent traits, as well as anankastic and schizoid traits, which strongly correlate with FFM linear composite in our study (N+, C-, A-, E-). In clinical settings, individuals with these traits often report interpersonal difficulties as a core of PD, experience significant distress, and struggle with adapting to their environment due to introversion and disorganization. These findings align with the expected relationships between PD factors and the FFM model, leading us to label the first quasi-canonical function as a “broad generalized personality disorder factor”. It is important to underscore that although this factor encompasses a broad spectrum of dimensions, the internalizing dimensions exhibit a more pronounced effect within this construct.

The second quasi-canonical function demonstrates the correlation between disinhibited/borderline externalizing and antisocial PD domains with extraversion (E+), openness (O+), and moderately low agreeableness (A-).

The PD set in the second quasi-canonical function found in our study represents the remaining end of the spectrum and predominantly represents the cluster B PDs (histrionic and narcissistic items).

If we analyze the contributions of normal traits separately, we need to examine the quasi-canonical synthetic variables belonging to the FFM set and analyze the structure of the FFM set. By doing so, we could connect our results with the findings of Strus et al. [27,28].

The linear composite of traits in our research (E+, O+, A-) is similar to the Delta-Minus/Sensation-Seeking metatraits (N+, E+, O+, A-, C-) in the CMP model, which is associated with impaired behavioral control, impulsivity, and affective instability. On the other hand, the opposite of the Delta-Plus/Self-Restrained metatrait is linked with conformity (N-, E-, O-, A+, C+) [27,28].

The second quasi-canonical function appears to be located between the Beta-Plus (plasticity) and Delta-Minus (sensation-seeking) metatraits in the CMP model. This function includes both adaptive and maladaptive traits, such as a willingness to engage in new experiences, a tendency to initiate social relations personally, and expansiveness in interpersonal relationships. However, it also encompasses impulsiveness, stimulation-seeking, and other maladaptive traits [47]. 

Our findings are supportive of the theoretically expected relationships between PD factors and the FFM model, and we labeled the second quasi-canonical function as an externalizing factor 

Redundancy analysis showed that there were minor differences in the amount of variance explained by each set. These differences were not significant enough to suggest that either set of variables explained a significantly higher proportion of variance in the other set. In other words, there is a reciprocal relationship between the personality traits identified in FFM and the dimensions of PD. Each set of traits helps to explain and define the other, creating a dynamic interplay between the two.

The internalizing and externalizing framework was initially utilized to classify child and adolescent clinical categories but has since been extended to adult psychiatry and as a framework for understanding personality pathology. This model has proven to be useful in explaining behavior, comorbidity, and developing therapeutic strategies [46]. In fact, some experts have suggested that future editions of the DSM might organize disorders into internalizing and externalizing sections [48].

A national epidemiological study found that PD such as antisocial, histrionic, and narcissistic were associated with the externalizing spectrum, while schizotypal, avoidant, and obsessive-compulsive personality disorders were associated with the internalizing spectrum. Borderline personality disorder was found to be connected with both spectra. Additionally, an overarching latent factor representing general personality dysfunction was significantly greater on the internalizing spectrum, which is consistent with the findings of our study [49].

These findings provide further support for the internalizing and externalizing framework and suggest that these dimensions play a crucial role in understanding the structure of psychopathology and developing targeted treatment strategies. The differentiation between treatment-seeking (Type S) and treatment-rejecting (Type R) PD is crucial, as interventions will differ based on the type of individual. In the case of Type S, the intervention will be directed towards the individual, whereas in the case of Type R, the intervention will focus on the environment through nidotherapy and social prescribing [50,51]. Tyrer et al. demonstrated that Cluster C personality disorders are more likely to be classified as Type S [51].

A study on a non-clinical population showed that BPD, together with cluster C and schizoid PD, belongs to Type S, while antisocial, histrionic, and narcissistic PDs belong to Type R. This paper demonstrated that Type S, compared to Type R, is strongly associated with high neuroticism and lower, but not statistically significant, agreeableness [52]. Our first quasi-canonical function in the PD set corresponds to these findings. Specifically, the borderline internalizing factor includes items related to avoidant, dependent (cluster C), and borderline PDs, as well as detachment items. Additionally, the FFM set is characterized primarily by high neuroticism and low agreeableness, consistent with the observed strong association between Type S and high neuroticism and lower but not statistically significant agreeableness [53].

Additionally, VanBeek and Verheul discovered that a crucial factor that predicts a person’s motivation for treatment is the extent of their distress symptoms [54]. Therefore, in our study, individuals with high levels of neuroticism, which correspond to the first function, would be more likely to seek treatment. In contrast, the second quasi-canonical function, consisting of antisocial, narcissistic, and histrionic PDs, would belong to type R, where symptoms are more ego-syntonic.

New research suggests that psychotherapies such as SFT and DBT, originally developed for the categorical model, may also benefit the dimensional model of PD. SFT constructs were particularly linked to personality traits, revealing consistent associations between AMPD and ICD-11 traits and specific maladaptive schemas [55]. Maladaptive coping was identified as a key determinant of PDs globally, being independently associated with all domains in both the AMPD and ICD-11. DBT could therefore be a suitable approach to target maladaptive coping in individuals with elevated traits in either model [52]. This highlights the potential utility of these therapies beyond their original scope for a wider range of personality disorders in a dimensional model.

## 5. Limitation

Our study included patients with PD and acute symptom exacerbation requiring intensive treatment. However, the sample may not be representative of all PD patients, as there may be a bias towards more severe and low-functioning PDs. This could be due to the fact that emotionally unstable PDs are more likely to require hospitalization. 

Additionally, the low representation of antisocial personality disorder in our sample may be due to its association with aggression and non-cooperation with voluntary treatment. Furthermore, excluding patients with acute psychotic disorders may have led to the underrepresentation of schizotypal personality disorder, which is often associated with psychotic decompensation and seeking treatment.

In terms of limitations, it is worth noting that the study did not incorporate a control group. Incorporating a control group could have shed light on the association between the SCID-5 and NEO-PI-R scales in a healthy population and would have been beneficial for interpreting the data.

In our study, we did not consider psychoticism as a feature because it is not included in the ICD-11 classification. These features are coded within the section on schizophrenia and other primary psychotic disorders. Although the ICD-11 approach does evaluate the capacity for reality testing when assessing the severity of PDs, it does not recognize psychoticism as a distinct domain. To more precisely capture the concept of psychoticism as a separate PD feature, we suggest that future studies incorporate additional assessment scales, such as the recently proposed broad, trait-like reconceptualization of proneness to psychotic-like experiences/behaviors.

## 6. Conclusions

Our study provides evidence for a joint structure of maladaptive and normal personality traits as well as the manner in which these structures are integrated. Our findings showed a relationship between the internalizing PD spectrum (consisting of avoidant, dependent, and borderline traits with detached and anankastic traits) and high neuroticism, low conscientiousness, and moderately low agreeableness, suggesting the existence of a broad personality disorder factor. However, the internalizing dimensions exhibited a more pronounced effect within this construct. Furthermore, we identified a second subgroup that demonstrated a link between the externalizing PD spectrum (including narcissistic, histrionic, and antisocial traits) and high extraversion, high openness, and low agreeableness, suggesting the existence of an externalizing factor that potentially moderates behavioral control. Our redundancy analysis revealed a reciprocal relationship between the personality traits identified in the FFM and the dimensions of PD. Each set of traits helps to explain and define the other, creating a dynamic interplay between the two. Overall, our findings emphasize the importance of integrating the FFM model into PD evaluation in clinical practice and suggest that differentiating between major subgroups could assist in adjusting therapeutic approaches for optimal outcomes.

## Figures and Tables

**Table 1 brainsci-13-00605-t001:** Correlations between extracted PD domain factors in previous study and FFM factors.

	Neuroticism	Extraversion	Openness	Agreeableness	Conscientiousness
Borderline internalizing	0.77 **	−0.35 **	−0.18 ^**^	−0.25 **	−0.51 **
Disinhibited/borderline externalizing	0.25 **	0.34 **	0.17 *	−0.44 **	−0.25 **
Antisocial	0.14 *	0.04	0.04	−0.31 **	−0.23 **
Anankastic	0.42 **	−0.19 **	−0.05	−0.27 **	−0.07
Detached	0.34 **	−0.42 **	−0.23 **	−0.17 *	−0.25 **

Notations: * *p* < 0.05; ** *p* < 0.01.

**Table 2 brainsci-13-00605-t002:** Function coefficients and structure coefficients for the first quasi-canonical function.

Variable Set 1 (PD Set)	Function Coefficients *	Structure Coefficients **	Variable Set 2 (FFM Set)	Function Coefficients	Structure Coefficients
Borderline internalizing	0.75	0.87	Neuroticism	0.71	0.88
Disinhibited/borderline externalizing	0.25	0.46	Extraversion	−0.3	−0.55
Antisocial	0.23	0.39	Openness	−0.14	−0.28
Anankastic	0.43	0.66	Agreeableness	−0.4	−0.47
Detached	0.37	0.66	Conscientiousness	−0.46	−0.76

* A structure coefficient represents the correlation between an observed variable and a synthetic variable. ** Standardized canonical function coefficients are the standardized coefficients used in linear equations to combine the observed set variables into two synthetic variables.

**Table 3 brainsci-13-00605-t003:** Function coefficients and structure coefficients for the second quasi-canonical function.

Variable Set 1 (PD Set)	Function Coefficients	Structure Coefficients	Variable Set 2 (FFM Set)	Function Coefficients	Structure Coefficients
Borderline internalizing	−0.17	−0.03	Neuroticism	0.00	−0.07
Disinhibited/borderline externalizing	0.84	0.85	Extraversion	0.74	0.8
Antisocial	0.35	0.51	Openness	0.39	0.63
Anankastic	−0.38	−0.34	Agreeableness	−0.53	−0.45
Detached	0.01	0.03	Conscientiousness	−0.15	−0.04

## Data Availability

Data supporting reported results are available from the corresponding author on request.

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
