# Peer review of "Analysis of the Relationship between Higher-Order Factor Structure of Personality Disorders and the Five-Factor Model of Personality"

_brainsci, 2023, doi:10.3390/brainsci13040605_

Round 1
Reviewer 1 Report
Comments and Suggestions for Authors
This paper tried to identify personality disorder clusters by analyzing higher-order factors derived from both personality disorder and five-factor models. However, there are some pitfalls in the article, as is shown below:
Major comments:
1. In the Abstract part, the authors mentioned the trend of using dimensional classification of personality disorders in ICD-11 based on the growing body of evidence. While in this study they tried to identify PD clusters and found two subgroups of PD, “borderline-internalizing” and “disinhibited/borderline externalizing” and antisocial ones, each of which include both emotional and interpersonal functioning and seem to comprise two continuities. So, it was confusing which classification method did the authors support? The categorical one or the dimensional one?
2. In the Introduction part, the authors mentioned that their current work was based on their previous one which found a two-factor solution after cross-validation of the ICD-11 domains in PD patients. However, the two factor structure hasn’t cover any Cluster A PD type, even the schizotypal type which mentioned in AMPD of DSM-5. The comprehensiveness and validation of the structure should be further evidenced and introduced. I have such concern that does it has something to do with the lower robustness of the other three factors (Antisocial, Anankastic, and Detached).
3. Previous work on the correlation between pathological personality traits and FFM one were not adequately introduced. For example, the findings of Wei Wang et al.,2003, BMC Psychiatry; Fan et al., 2016, BMC Psychiatry; etc.
4. In the Method part, the authors mentioned that they mainly enrolled PD patients comorbid with affective disorders, which might lead to the skewness of participant enrollment with more obviosity on the “borderline” domain, and influence the identification of higher-order PD factors.
5. The possible application of this work in clinic should be further explained.
Minor comments:
6. The titles of Table 2 and Table 3 which used the “first quasi-canonical function” and “second quasi-canonical function” didn’t match the narrations in the Results part. For example, in the 3rd and 4th paragraphs of Results, the authors mentioned “Second quasi-canonical functions…”, “Function 1”, which would be better unified.
7. It would be better if the authors could explain the meaning of “function coefficients” and “structure coefficients” in Table 2 and Table 3 with a note below.
Author Response
Response to Reviewer 1 Comments
Point 1: In the Abstract part, the authors mentioned the trend of using dimensional classification of personality disorders in ICD-11 based on the growing body of evidence. While in this study they tried to identify PD clusters and found two subgroups of PD, “borderline-internalizing” and “disinhibited/borderline externalizing” and antisocial ones, each of which include both emotional and interpersonal functioning and seem to comprise two continuities. So, it was confusing which classification method did the authors support? The categorical one or the dimensional one?
Response 1: Thank you for your comment on our paper. As authors, we would like to clarify that we strongly support dimensional classification of personality disorders. We believe that a dimensional approach can provide a more nuanced understanding of personality disorders and better inform treatment and interventions.
One of the authors of our paper, Prof. Dusica Lecic Tosevski, is also a member of the working group for personality disorders of the ICD-11 classification. Our work is based on previously published research where we examined the factor structure of personality disorder domains using the methodology of the ICD-11 working group and applied factor analysis to the items of the DSM model (Mulder et al.). We acknowledge that factors or domains can be seen as dimensions, which supports the use of dimensional classification.
We will ensure to clearly state our support for dimensional classification in the paper, specifically in the introduction or conclusion sections, to avoid any confusion or misinterpretation of our findings in relation to classification methods.
Point 2: In the Introduction part, the authors mentioned that their current work was based on their previous one which found a two-factor solution after cross-validation of the ICD-11 domains in PD patients. However, the two factor structure hasn’t cover any Cluster A PD type, even the schizotypal type which mentioned in AMPD of DSM-5. The comprehensiveness and validation of the structure should be further evidenced and introduced. I have such concern that does it has something to do with the lower robustness of the other three factors (Antisocial, Anankastic, and Detached).
Response 2: We appreciate your concern about the comprehensiveness and validation of the structure we proposed, particularly with regards to the absence of Cluster A PD types in our proposed structure.
As we stated in our abstract, our previous work validated the proposed five-factor structure of the ICD-11 domain to some extent. The first two factors, labeled as borderline-internalizing and disinhibited/borderline externalizing, were the most robust. However, the other three factors, labeled as antisocial, anankastic, and detached, were less robust.
Regarding the absence of Cluster A PD types in our proposed structure, we agree that this raises questions about its comprehensiveness. We acknowledge that excluding patients with acute psychotic disorder, who may be more likely to have schizotypal PD, may have contributed to the underrepresentation of schizotypal PD in our sample. However, it's worth noting that some paranoid items were included in the borderline-internalizing factor, and schizoid items formed a detached factor, suggesting that aspects of Cluster A may still be present in our proposed structure.
Point3: Previous work on the correlation between pathological personality traits and FFM one were not adequately introduced. For example, the findings of Wei Wang et al.,2003, BMC Psychiatry; Fan et al., 2016, BMC Psychiatry; etc.
Response 3: Thank you for bringing to my attention the significant work of the Chinese author on cultural specificity in the diagnosis and treatment of personality disorders. To further enrich our discussion or provide an informative introduction, we could summarize the article or cite it as a reference.
“The study aimed to explore the potential link between the Chinese Adjective Descriptors of Personality (CADP) and different personality disorder functioning styles. CADP measures the normal personality traits of Intelligent, Emotional, Conscientious, Unsocial, and Agreeable, which are comparable to the five-factor model of personality traits. The researchers investigated this association in both personality disorder patients and healthy volunteers. The results revealed that all five CADP traits were significantly linked to almost all 11 personality disorder functioning styles in patients. Additionally, personality disorder patients scored higher on the CADP Emotional and Unsocial traits than healthy volunteers” (Fan H, Zhu Q, Ma G, Shen C, Zhang B, Wang W. Predicting personality disorder functioning styles by the Chinese Adjective Descriptors of Personality: a preliminary trial in healthy people and personality disorder patients. BMC Psychiatry. 2016 Aug 30;16(1):302. doi: 10.1186/s12888-016-1017-0. PMID: 27578005;)
We will also cite this work.
Wang W, Hu L, Mu L, Chen D, Song Q, Zhou M, Zhang W, Hou J, Li Z, Wang J, Liu J, He C. Functioning styles of personality disorders and five-factor normal personality traits: a correlation study in Chinese students. BMC Psychiatry. 2003 Sep 17;3:11. doi: 10.1186/1471-244X-3-11. Epub 2003 Sep 17
Point4: In the Method part, the authors mentioned that they mainly enrolled PD patients comorbid with affective disorders, which might lead to the skewness of participant enrollment with more obviosity on the “borderline” domain, and influence the identification of higher-order PD factors.
Response 4: Thank you for your important comment. As you pointed out, our study enrolled participants with comorbid affective disorders, with mood disorders being the most prevalent (58.2%). This could have led to a skewed enrollment of individuals with borderline personality traits and may have influenced the identification of higher-order personality disorder factors.
Our finding is consistent with existing literature that reports anxiety and mood disorders as the most common comorbidities in patients with personality disorders. Additionally, major depression was a significant reason for intensive treatment in samples of hospital-treated patients.
To minimize this potential bias, our future studies could consider wider sampling and inclusion of patients receiving ambulatory treatment or psychotherapy.
Point5: The possible application of this work in clinic should be further explained.
Response 5: Thank you for your suggestion. We have now included a paragraph in our study discussing the connection between dimensional classification and specialized therapy for personality disorders, including schema-focused therapy and dialectical behavior therapy.
Minor comments:
Point6. The titles of Table 2 and Table 3 which used the “first quasi-canonical function” and “second quasi-canonical function” didn’t match the narrations in the Results part. For example, in the 3rd and 4th paragraphs of Results, the authors mentioned “Second quasi-canonical functions…”, “Function 1”, which would be better unified.
Response 6: Thank you. After a thorough review, we have harmonized the terminology used throughout the work, particularly with regards to the methodology and use “first quasi-canonical function” and “second quasi-canonical function”
Point7. It would be better if the authors could explain the meaning of “function coefficients” and “structure coefficients” in Table 2 and Table 3 with a note below.
Response 7: Thank yo for your suggestion. We included comments about coefficients and its meaning, below the tables 2 and 3.
Reviewer 2 Report
Comments and Suggestions for Authors
Review Brain Sciences (ISSN 2076-3425) Comprehensive Analysis of the Relationship between Higher-Order Factor Structure of Personality Disorders and the Five-Factor Model of Personality
Interesting structural study on components of higher order factors in PD and FFM.
The title “comprehensive” doesn’t mean anything really? It is more of a factorial analysis (extracting latent factors).. You could even argue it is a few simple correlations which doesn’t suffice for an entire publication.
The abstract summarizes a broad factor of PD and a thrill-seeking factor. In the Hi-TOP research and others these are obviously internalizing-externalizing – Why use different labels, only potentially confusing the field?
Also, Figure 3 in the following reference, may be of interest and worth comparing with in the discussion? Kajonius, P. J., & Dåderman, A. M. (2017). Conceptualizations of personality disorders with the five factor model-count. International Journal of Testing, 17(2), 141-157.
Seeing how the introduction lays the foundation of the FFM in relation to PD, shouldn’t Miller and Lynam et al.’s research for the last twenty years be referenced. They have done work relevant to the topic of the present study. Ex. Miller, J. D., Bagby, R. M., Pilkonis, P. A., Reynolds, S. K., & Lynam, D. R. (2005). A simplified technique for scoring DSM-IV personality disorders with the five-factor model. Assessment, 12(4), 404-415.
The Figure 1 and canonical analysis is not unlike the approach by the Figure in Kajonius, P. J. (2017). The short personality inventory for DSM-5 and its conjoined structure with the common five-factor model. International Journal of Testing, 17(4), 372-384.
Useful sample. Remember though that 223 persons don’t make a power-driven study, especially not seeing the factor analytical approaches.
The study design/analysis section doesn’t hold up. It is virtually impossible for the reader to replicate the study based on this. Mostly there is information about the methods, not how ICD-instruments form factors, on what basis extraction has been done, what extraction methods, what limits of “robustness” are, the reader basically only can join for the ‘synthetic correlations’. It is simply not sufficient nor readable enough, sorry.
It is questionable whether EFA should even be used for extracting domains of PD. EFA is allowed to cross-load freely and error-control is dubious compared to CFA, especially based on a single sample like the present. I am not sure the present study is so useful to the field seeing these extracted variables cannot be compared to the full extent to the rest of the body of research?
Why not merge Table 1 and 2?
Why use three decimals in a sample of 223?
Agreeableness (or similar) tends to be a powerful component of all relational PD’s – This doesn’t show in the present study?
Refering to the “previous study” doesn’t work very well in my opinion. Use straight formal reference instead. Also, I the reader doesn’t find that study convincing – on what authority are the other ICD-11 domains not “robust”? Does that mean we should only use the extracted ones, and does that imply to overthrow decades worth of work on putting together the new ICD-11 (and future DSM-5)?
The discussion contains many many threads, labels, concepts, models, and is really hard to make out. It is like attempting to touch upon everything, which in my opinion only makes the point weaker.
Overall, I enjoy the attempt of explaining PD and FFM and I think it is an important endevour. However, as an informed reader following the present manuscript I have a really hard time placing the present study result. How does it fit with DSM-5, Hi-TOP, FFM-count, P-factor, G-factor. None of this seems to be obvious, and if not, what does the present study add or prove?
Thanks for letting me read the paper. I encourage you to attempt much more clarity, simplicity, transparency. Why not compare several higher order factor-models and graphically present the actual instruments used (instead of a general hard to see Figure 1), for instance.
Author Response
Response to Reviewer 2 Comments
Point 1: The title “comprehensive” doesn’t mean anything really? It is more of a factorial analysis (extracting latent factors).. You could even argue it is a few simple correlations which doesn’t suffice for an entire publication.
Response 1:
We appreciate your feedback regarding the title of our paper. We acknowledge that the use of the word 'comprehensive' may be pretentious and could have been better phrased. Our intention was to convey the complexity of our analysis, as we aimed to explore the relationship between the higher-order factors of personality disorders and the domains of the five-factor personality model, building on our previous work in which we attempted to cross-validate the proposed five factors for the ICD-11 classification. To achieve this, we employed canonical analysis of covariance, a powerful statistical analysis method. Nonetheless, we will revise the title to better reflect the focus and scope of our study, the word "comprehensive" may not be necessary.
Point 2: The abstract summarizes a broad factor of PD and a thrill-seeking factor. In the Hi-TOP research and others these are obviously internalizing-externalizing – Why use different labels, only potentially confusing the field?
Response 2: Thank you very much for this important question. As a group, we hold a different attitude towards names because we recognize that although the current domain proposals were accepted as final, the content and domain names are constantly being reevaluated and changed. For example, in one of the latest revalidations of the proposition by Mulder et al., the domain named "Borderline" consisted of borderline, histrionic, and narcissistic symptoms, while the "Disinhibited" domain was not a distinct domain but loaded onto a factor named "Dissocial/Disinhibited." In the current proposal, these factors are separated, and the Borderline pattern has disappeared and been later accepted as "Borderline pattern."
However, since we are dealing here with factors of a higher order, we agree that it is more appropriate to call the second factor the Externalizing factor to avoid any additional confusion.
Point3: Also, Figure 3 in the following reference, may be of interest and worth comparing with in the discussion? Kajonius, P. J., & Dåderman, A. M. (2017). Conceptualizations of personality disorders with the five factor model-count. International Journal of Testing, 17(2), 141-157.
Response 3: Thank you for referring us to the works of Petri Kanonijus. In the discussion, we can connect the results of his work with our results.
One objective of Kajonijus and Daderman's study was to assess dispositions for DSM categories based on normal personality continuums. To achieve this, they linked personality disorder (PD) categories to normal personality traits on a continuum. The FFM-count method was used for PD scores, which relies on expert prototypes that assign each PD category a distinct set of FFM facets.To generate higher-order variables, the researchers employed a "bass-ackwards" approach, which allowed for a top-down structure with path coefficients. This was achieved by developing a hierarchical factor structure level by level, starting with the extraction of one fixed component and progressing to two, three, and so on, while correlating the components between levels. In the end, they chose to utilize exploratory factor analysis (EFA) due to the known comorbidity between factors. Higher-order factors derived from the factor analyses were found to align well with previously established higher-order organizational frameworks, such as internalizing and externalizing.
Point4: Seeing how the introduction lays the foundation of the FFM in relation to PD, shouldn’t Miller and Lynam et al.’s research for the last twenty years be referenced. They have done work relevant to the topic of the present study. Ex. Miller, J. D., Bagby, R. M., Pilkonis, P. A., Reynolds, S. K., & Lynam, D. R. (2005). A simplified technique for scoring DSM-IV personality disorders with the five-factor model. Assessment, 12(4), 404-415.
Response 4: Thank you for the important suggestion. In the introduction, we cited Miller and
Lynam's important contribution to the understanding of the relationship between the FFM model and DSM personality pathology.
Point5: The Figure 1 and canonical analysis is not unlike the approach by the Figure in Kajonius, P. J. (2017). The short personality inventory for DSM-5 and its conjoined structure with the common five-factor model. International Journal of Testing, 17(4), 372-384.
Response 5: Thank you very much for bringing up the work of Kajonijus in this article. We have noticed the similarity between our work and theirs, and we have discussed the conjoint structure of DSM and FFM with our results. We appreciate your input and the opportunity to further expand our understanding of this topic.
Point6: Useful sample. Remember though that 223 persons don’t make a power-driven study, especially not seeing the factor analytical approaches.
Response 6: Thank you for acknowledging the limitations of our study, particularly in regards to the small sample size used in the factor analytical study conducted in our previous research. It is important to recognize that a small sample size may limit the generalizability and reliability of the findings.
Point 7: The study design/analysis section doesn’t hold up. It is virtually impossible for the reader
to replicate the study based on this. Mostly there is information about the methods, not how ICD-instruments form factors, on what basis extraction has been done, what extraction methods, what limits of “robustness” are, the reader basically only can join for the ‘synthetic correlations’. It is simply not sufficient nor readable enough, sorry.
Response 7: Thank you very much for your note. It was the subject of a debate among our group of authors, and we have decided to recount our previous methodology only to some extent because (“We applied the same framework, with a slightly smaller number of item parcels and a different grouping of the 57 DSM symptom criteria in a sample of PD patients. To analyze the data, we used exploratory factor analysis with principal axis factoring as the extraction method and Promax as the oblique rotation method”) We did not show how we choose items and created item parcels because it is explained in detail in our previous work.
We have now included more concrete data about EFA in our new manuscript (“Exploratory factor analysis (EFA) was conducted on 21 item parcels, with a ratio of 10.6 cases per variable. The Kaiser-Meyer-Olkin measure of sampling adequacy was 0.81, and Bartlett’s test of sphericity was significant (χ2 (210) = 1350.62, p < 0.01), indicating that the data were appropriate for factor analysis. We extracted five factors with eigenvalues greater than 1, which collectively accounted for 41.39% of the variance. Factor score estimates were generated for each of the five factors by summing the individual items within each item parcel that loaded onto the respective factor, and factor reliabilities were assessed using Cronbach's alpha. Reliability estimates were strong for the borderline-internalizing (α = 0.86) and antisocial (α = 0.80) factors, acceptable for the disinhibited/borderline externalizing factor (α = 0.71), and poor for the anankastic (α = 0.55) and detached (α = 0.53) factors.”)
Thank you for your suggestion. We have also made a straightforward reference to our previous work, as you suggested in your points.
Point8: It is questionable whether EFA should even be used for extracting domains of PD. EFA is allowed to cross-load freely and error-control is dubious compared to CFA, especially based on a single sample like the present. I am not sure the present study is so useful to the field seeing these extracted variables cannot be compared to the full extent to the rest of the body of research?
Response 8: Thank you for raising this most important point. We want to clarify that our choice to use EFA instead of CFA was also a topic of debate with the editor in our previous journal submission, and we have added an explanation of our reasoning in the limitations section (Pesic D, Lecic-Tosevski D, Kalanj M, Vukovic O, Mitkovic-Voncina M, Peljto A, Mulder R. Multiple faces of personality domains: Revalidating the proposed domains. Psychiatr Danub. 2019 Jun;31(2):182-188).
Due to the size of our sample, we had a relatively low prevalence of symptoms for some PDs, specifically antisocial and schizoid symptoms.
Our primary goal was to use the same framework of grouping and choosing DSM items and to perform the same procedure of grouping, but on a different population, specifically a sample of primary PD patients. We included the same framework and 57 DSM-IV symptom criteria as in the previous study by Malder et al.
In the limitations section in previous article, we stated that "We applied only EFA and did not test the five-factor model using confirmatory factor analysis (CFA). The recommendation in the literature is to perform both an EFA on half of the sample and a CFA on the other half when cross-validating results of factor studies (Brown, 2006). We decided to apply only EFA due to limitations imposed by the number of participants, as splitting our sample in half would reduce the number of cases per variable ratio." We appreciate your input and hope this clarifies our reasoning.
Point 9: Why not merge Table 1 and 2?
Response 9: Thank you for your suggestion. To clarify, I understand that you may have suggested combining tables 2 and 3, which contained the function coefficients and structure coefficients for the first and second quasi-canonical function. Becasue it is separate orthogonal functions we presented it separately. Technically, the tables are large, we tried it and it seems very cluttered in the format where the tables are in text. Of course we can try to find a solution if it is technically feasible
Point 10: Why use three decimals in a sample of 223?
Response 10: Thank you for your suggestion. We agree that reducing the number of decimal places in the tables and changing their presentation.We will implement these changes in revisited version of the manuscript.
Point 11: Agreeableness (or similar) tends to be a powerful component of all relational PD’s – This doesn’t show in the present study?
Response 11: Thank you for the opportunity to discuss the importance of agreeableness in our sample. It's worth noting that low agreeableness is present in both functions, as evidenced by the results presented in Table 1 and Table 2.
In the first function, the second synthetic variable accounted for 39.2% of the variance and included neuroticism (N+), negative conscientiousness (C-), and negative agreeableness (A-), with a structure coefficient of -0.475 for agreeableness.
In the second function, the second synthetic variable in the FFM set explained 24.9% of the variance and was characterized by positive extraversion (E+), positive openness (O+), and moderately negative agreeableness (A-), with a structure coefficient of -0.455 for agreeableness.
Overall, these findings highlight the importance of agreeableness in our analysis, and we did not said that in conclussion which was changed.
Point 12: Refering to the “previous study” doesn’t work very well in my opinion. Use straight formal reference instead. Also, I the reader doesn’t find that study convincing – on what authority are the other ICD-11 domains not “robust”? Does that mean we should only use the extracted ones, and does that imply to overthrow decades worth of work on putting together the new ICD-11 (and future DSM-5)?
Response 12: Thank you very much for this important note. Perhaps we did not emphasize our previous study enough, but we made a straight formal reference to it under number 14: Pesic D, Lecic-Tosevski D, Kalanj M, Vukovic O, Mitkovic-Voncina M, Peljto A, Mulder R. Multiple faces of personality domains: Revalidating the proposed domains. Psychiatr Danub. 2019 Jun;31(2):182-188. doi: 10.24869/psyd.2019.182. The main objective of our study was to cross-validate the mode used by Mulder et al. (one of the authors in our article) in order to validate the factor structure of personality disorder symptoms in a different sample of patients diagnosed primarily with personality disorders. To achieve this, we included the same 57 DSM-IV symptom criteria as Mulder et al. did in their study. Their study aimed to validate the proposed ICD-11 domains (Mulder RT, Horwood J, Tyrer P, Carter J, Joyce PR. Validating the proposed ICD-11 domains. Personal Ment Health. 2016 May;10(2):84-95. doi: 10.1002/pmh.1336. PMID: 27120419).
In our previous study, we also discovered dissocial, detached, and anankastic domains that closely resembled those identified in Mulder et al.'s (2016) study. However, we stated that in our sample, the evidence for these domains was not very robust (not in ICD-11) The antisocial factor accounted for only a small portion of variance (around 5%), but this may be due to the low prevalence of Antisocial PD in our sample (1.5%). Nonetheless, it was clearly distinguished as a separate factor with satisfactory estimated reliability, and it was moderately correlated with the externalizing factor. We also encountered a specific issue with the low reliability of schizotypal items, which led to weak evidence of a detached factor based only on schizoid items. Obsessive-compulsive symptoms were observed to load significantly onto the anankastic factor, but this factor accounted for only a small portion of variance, despite the high prevalence of obsessive-compulsive symptoms in our sample. Moreover, it had poor estimated reliability and was strongly correlated with the "borderline-internalizing" factor.
Point 13: The discussion contains many many threads, labels, concepts, models, and is really hard to make out. It is like attempting to touch upon everything, which in my opinion only makes the point weaker.
Response 13: In response to your comment, we have revised our manuscript to narrow the scope of the discussion by excluding sections that were not directly relevant to our results. Thank you for your valuable feedback.
Point 14: Overall, I enjoy the attempt of explaining PD and FFM and I think it is an important endevour. However, as an informed reader following the present manuscript I have a really hard time placing the present study result. How does it fit with DSM-5, Hi-TOP, FFM-count, P-factor, G-factor. None of this seems to be obvious, and if not, what does the present study add or prove?
Response 14: Thank you for your comments and question. Our results indicated that the first function explained the majority of variance, incorporating higher-order factors from previous research, such as Borderline Internalizing (Borderline, Dependent and Avoidant), Detached, and Anankastic, as well as typical FFM dimensions like N+, C-, and A-. These factors corresponded to a broad disturbance in personality. It is important to underscore that although this factor encompasses a broad spectrum of dimensions, the internalizing dimensions exhibit a more pronounced effect within this construct
On the other hand, the second function explained the remaining variance and it connected the Disinhibited/Borderline externalizing factor (Narcissistic and Histrionic items) and the Antisocial factor with high extraversion(E+) and low agreeableness (A-). This second function (externalisation) demonstrated that there is a moderating element that modifies the general personality disorder factor.
We used CAC as the parent linear general model to investigate the relationship between two interrelated variables, SCID and NEO, and determine the variables that accounted for the highest degree of shared variance.
We rewrite that “Our main research question was focused on the extent and manner in which maladaptive and common trait structures are integrated. We hypothesized that there would be evidence of a mutual structure of a latent higher-order factor”
We tried to be clearer and more precise in the revised version
Point 15: Thanks for letting me read the paper. I encourage you to attempt much more clarity, simplicity, transparency. Why not compare several higher order factor-models and graphically present the actual instruments used (instead of a general hard to see Figure 1), for instance.
Response 15:
Thank you very much for your valuable feedback and suggestion. We appreciate your input and agree that exploring the connection of FFM scales with other factor solutions would be a valuable direction for future research. In this study, our aim was to explore the connection between the FFM scale and a previously published study of the personality disorder factor. The primary attempt of the previous study was very simple, only to cross-validate Mulder's study but in a different sample of patients with personality disorder. Therefore, we used our previously published results and aimed to extend those findings. We will definitely take your suggestion into consideration for our future work.
Reviewer 3 Report
Comments and Suggestions for Authors
Dear authors,
I would like to thank you for the opportunity to review this manuscript. There are my comments:
Introduction: I have no remarks for on the Introduction section.
Material and methods:
1. To add at the exclusion criteria: severe depressive episode, maniacal episode, bipolar disorder
2. I don't see the necessity to figure the schematic presentation of the principles of Canonical Correlation Analysis while it isn't the statistical method used in this article.
Results:
It is necessary to explain the following terms:
- function/structure coefficients (theoretically as well as the relevant values in Table 2 and Table 3)
- 40.4% of the variance
- 19.5% of the variance...19.1% of the variance
- 21.8% of the variance...24.9% of the variance
Discussion:
Re-write this section for better ordering (the content is relevant but could be more explicit if the comparative data would be better structured).
Conclusions:
Expand '"the internalized PD spectrum", "the externalized PD spectrum" and their meaning in the theorethical concept of PD.
Author Response
Response to Reviewer 3 Comments
Point 1: To add at the exclusion criteria: severe depressive episode, maniacal episode, bipolar
disorder
Response 1: Thank you for your valuable suggestion. We have expanded the exclusion criteria as per your recommendation to enhance the study's validity.
Point 2: I don't see the necessity to figure the schematic presentation of the principles of Canonical Correlation Analysis while it isn't the statistical method used in this article
Response 2: Thank you for your helpful suggestion. Our initial aim was to provide a schematic presentation of canonical analysis of covariance through canonical correlation analysis. However, we agree that this may have caused confusion for the reader and have therefore decided to remove this figure. Thank you for bringing this to our attention and for your valuable input.
Point 3: It is necessary to explain the following terms:
- function/structure coefficients (theoretically as well as the relevant values in Table 2 and Table 3)
- 40.4% of the variance 39
- 19.5% of the variance...19.1% of the variance
- 21.8% of the variance...24.9% of the variance
Response 3: Thank you for bringing up an important point. We have included an explanation in our report regarding the difference in coefficients between the theoretical model and our study results. Furthermore, we have addressed the difference in variance between the two functions.
-The quasi-canonical correlation coefficient is the correlation between the two synthetic variables on a given quasi-canonical function
-A structure coefficient is the correlation between an observed variable and a synthetic variable. In CAC, it is the correlation between an observed variable and the canonical function scores for the variable’s set (that is, the synthetic variable created from all the set variables by their linear combination).
-Standardized canonical function coefficients are the standardized coefficients used in the linear equations to combine the observed set variables into two synthetic variables.
-A canonical communality coefficient is the proportion of variance in each variable that is explained across the canonical solution being interpreted.
We improved the clarity of the results and discussion sections of a research paper.
“The results of the quasi-canonical correlation analysis indicate the presence of two quasi-canonical functions, each comprising pairs of quasi-canonical synthetic variables belonging to the PD set and FFM set, respectively.
We showed that there were variances in the first quasi-canonical function for two sets, PD and FFM, at 40% and 39% respectively. Similarly, we observed variances in the second quasi-canonical function for these two sets at 21.8% and 24.9%, respectively.
We conducted redundancy analysis to explore the relationship between the two sets of variables. The first quasi-canonical function explained 19.5% of the variance in PD variables by the five-factor personality FFM measures, and 19.1% of the variance in the five-factor personality variables by the PD variable set. For the second quasi-canonical function, 8.4% of the variance in PD variables was explained by the five-factor personality FFM measures, and 9.4% of the variance in the five-factor personality variables was explained by the PD variable set.
In disscussion we explained meaning of redundancy analysis.
Redundancy analysis showed that there were minor differences in the amount of variance explained by each set. These differences were not significant enough to suggest that either set of variables explained a significantly higher proportion of variance in the other set. In other words, there is a reciprocal relationship between the personality traits identified in FFM and the dimensions of PD. Each set of traits helps to explain and define the other, creating a dynamic interplay between the two.
Point 4: Discussion
Re-write this section for better ordering (the content is relevant but could be more explicit if the comparative data would be better structured).
Response 4: Thank you for your feedback. We have revised the discussion section to improve
clarity and make our conclusions more explicit.
Point 5: Expand '"the internalized PD spectrum", "the externalized PD spectrum" and their meaning in the theorethical concept of PD.
Response 5: Thank you for your comment. We have expanded the theoretical concept of the internalized and externalized personality disorder spectrum in our report to provide a more comprehensive understanding of these constructs. We appreciate your input and have worked to incorporate it into our report.
Reviewer 4 Report
Comments and Suggestions for Authors
This is a timely and scientifically interesting investigation of personality disorders and personality traits in more than 220 patients with personality disorders. The ICD-11 classification and the 5-factor-model of personality are the focus of the project and the article. I have only minor comments:
- The limitations section should mention that there was no control group. The authors mention that the investigated group of patients were severely affected by their personality disorder and that a group of less affected patients would have been helpful. However, it would be interesting to see how healthy people score on the both the SCID-5 and the NEO-PI-R and how both scales are associated in a healthy population. Additionally, the differences in scores between clinically diagnosed and people from the general population would be helpful in order to interpret the data.
- The authors claim that their work has clinical and practical implications. However, the readers will ask themselves what this might mean. I am questioning whether the research has practical psychotherapeutic consequences. Therefore, the authors should add a few sentences about this.
- Current therapies for personality disorders like transference-focused therapy, DBT or schema therapy should be mentioned in the introduction or discussion. They are to some degree built on diagnostic categories of personality disorders. There are other approaches like the use of music that might be more open to a spectrum model. It might be good to refer to those therapeutic options, cite the latest or most comprehensive reviews and explain how the diagnostic model might influence the therapeutic approach. One or two paragraphs should suffice.
Author Response
Response to Reviewer 4 Comments
Point 1: The limitations section should mention that there was no control group. The authors
mention that the investigated group of patients were severely affected by their personality disorder and that a group of less affected patients would have been helpful. However, it would be interesting to see how healthy people score on the both the SCID-5 and the NEO-PI-R and how both scales are associated in a healthy population. Additionally, the differences in scores between clinically diagnosed and people from the general population would be helpful in order to interpret the data.
Response 1: Thank you very much for your point. We put in limitation paragraph next part : “In terms of limitations, it is worth noting that the study did not incorporate a control group. Incorporating a control group could have shed light on the association between the SCID-5 and NEO-PI-R scales in a healthy population and would have been beneficial for interpreting the data”
Point 2: The authors claim that their work has clinical and practical implications. However, the readers will ask themselves what this might mean. I am questioning whether the research has practical psychotherapeutic consequences. Therefore, the authors should add a few sentences about this.
- Current therapies for personality disorders like transference-focused therapy, DBT or schema therapy should be mentioned in the introduction or discussion. They are to some degree built on diagnostic categories of personality disorders. There are other approaches like the use of music that might be more open to a spectrum model. It might be good to refer to those therapeutic options, cite the latest or most comprehensive reviews and explain how the diagnostic model might influence the therapeutic approach. One or two paragraphs should suffice
Response 2: Thank you for this important suggestion. We have included two paragraphs in this report that discuss the dimensional classification of personality disorders and the use of specialized psychotherapy for their treatment.
“New research suggests that psychotherapies like SFT and DBT, originally developed for the categorical model and primarily used to treat borderline personality disorder, may also benefit the dimensional model of PD. Specific schema domains were found to be associated with each personality trait domain, indicating trait domain-schema domain specificity. SFT constructs were particularly linked to personality traits, revealing consistent associations between AMPD and ICD-11 traits and specific maladaptive schemas. Maladaptive coping was identified as a key determinant of PDs globally, being independently associated with all domains in both the AMPD and ICD-11. DBT could therefore be a suitable approach to target maladaptive coping in individuals with elevated traits in either model. This highlights the potential utility of these therapies beyond their original scope for a wider range of personality disorders.”
Tracy M, Sharpe L, Bach B, Tiliopoulos N. Connecting DSM-5 and ICD-11 trait domains with schema therapy and dialectical behavior therapy constructs. Personal Ment Health. 2022 Dec 27 [Epub ahead of print]. doi: 10.1002/pmh.1574. PMID: 36575608
Bach B, Bernstein DP. Schema therapy conceptualization of personality functioning and traits in ICD-11 and DSM-5. Curr Opin Psychiatry. 2019 Jan;32(1):38-49. doi: 10.1097/YCO.0000000000000464. PMID: 30299307.
Round 2
Reviewer 1 Report
Comments and Suggestions for Authors
None.
Reviewer 2 Report
Comments and Suggestions for Authors
none